# Longitudinal study of *Chlamydia pecorum* in a healthy Swiss cattle population

**Samuel Loehrer**[1], **Fabian Hagenbuch**[1], **Hanna Marti**[1], **Theresa Pesch**[1], **Michael Hässig**[2], **Nicole Borel**[1] *

**1** Institute of Veterinary Pathology, Vetsuisse Faculty, University of Zurich, Zurich, Switzerland,
**2** Department for Farm Animals, Section for Herd Health, Vetsuisse Faculty, University of Zurich, Zurich, Switzerland

\* nicole.borel@uzh.ch

**Data Availability Statement:** All relevant data are within the paper and its Supporting Information files.

## Abstract

*Chlamydia pecorum* is a globally endemic livestock pathogen but prevalence data from Switzerland has so far been limited. The present longitudinal study aimed to get an insight into the *C. pecorum* prevalence in Swiss cattle and investigated infection dynamics. The study population consisted of a bovine herd (n = 308) located on a farm in the north-eastern part of Switzerland. The herd comprised dairy cows, beef cattle and calves all sampled up to five times over a one-year period. At each sampling timepoint, rectal and conjunctival swabs were collected resulting in 782 samples per sampled area (total n = 1564). *Chlamydiaceae* screening was performed initially, followed by *C. pecorum*-specific real-time qPCR on all samples. For *C. pecorum*-positive samples, bacterial loads were determined. In this study, *C. pecorum* was the only chlamydial species found. Animal prevalences were determined to be 5.2–11.4%, 38.1–61.5% and 55–100% in dairy cows, beef cattle and calves, respectively. In all categories, the number of *C. pecorum*-positive samples was higher in conjunctival (n = 151) compared to rectal samples (n = 65), however, the average rectal load was higher. At a younger age, the chlamydial prevalence and the mean bacterial loads were significantly higher. Of all sampled bovines, only 9.4% (29/308) were high shedders (number of copies per µl >1,000). Calves, which tested positive multiple times, either failed to eliminate the pathogen between sampling timepoints or were reinfected, whereas dairy cows were mostly only positive at one timepoint. In conclusion, *C. pecorum* was found in healthy Swiss cattle. Our observations suggested that infection takes place at an early age and immunity might develop over time. Although the gastrointestinal tract is supposed to be the main infection site, *C. pecorum* was not present in rectal samples from dairy cows.

## Introduction

*Chlamydia (C.) pecorum* belongs to the family *Chlamydiaceae* which currently contains 13 species [1]. Of these, four species have been reported to infect cattle, both as single and mixed infections: *C. abortus*, *C. pecorum*, *C. psittaci* and *C. suis* [1–4]. The most common chlamydial species prevalent in cattle are *C. pecorum* and *C. abortus*, both may cause sporadic abortions in

**Funding:** This study was supported by Seed Money AgroVet-Strickhof from the Vetsuisse Faculty and the Fondation sur la Croix.

**Competing interests:** The authors have declared that no competing interests exist.

bovines [5]. *C. psittaci* may induce respiratory signs in natural and experimental infections [6–9]. *C. suis* has only rarely been detected in bovines and its clinical relevance remains unclear [2, 10].

Like all *Chlamydiaceae*-family members, *C. pecorum* shares a characteristic, obligate intracellular biphasic life cycle preferentially replicating in epithelial cells. *C. pecorum* has a broad host range encompassing more than 20 animal species, including livestock (cattle, goats, sheep, pigs) and wild ruminants as well as exotic animals such as koalas [2, 11–14]. Until recently, it was assumed that *C. pecorum* has no zoonotic potential [15]. However, a case of severe community-acquired pneumonia has been reported in 2022, therefore requiring further monitoring [16]. In animals, the main site of infection is the gastrointestinal tract [12, 17], but *C. pecorum* can be also found in the urogenital and respiratory tract as well as in the conjunctiva and synovia [18, 19]. The main excretion route is by fecal shedding also occurring in an intermittent fashion [20] but vaginal, conjunctival, and nasal shedding has also been demonstrated [12, 21]. One study [8] suggests that venereal transmission is possible following the detection of *C. pecorum* in the semen of healthy bulls. However, the primary route of infection is likely feco-oral, aerogen or direct animal-animal contact [12, 22]. Although *C. pecorum* is responsible for significant economic losses in lambs due to polyarthritis, especially in Australia [22], the infection remains mostly asymptomatic in cattle [2, 20]. However, economic losses on a herd level due to *C. pecorum* have been reported and were a result of reduced growth rates, dairy production, or fertility [20, 23]. A study from the USA even showed a 48% reduction of growth in calves [24], however, the economic impact has been questioned in other studies [25, 26]. In addition to asymptomatic infection, *C. pecorum* can cause a wide range of clinical signs in bovines. Typically, bovines get infected at a young age and multiple times during their lifetime with different strains [6]. Several studies showed that *C. pecorum* is ubiquitous in livestock [12, 25, 27]. Prevalences of *C. pecorum* in dairy cows range between 0.7–8.9% in Europe [28, 29], which is similar to the USA (7.5%) [6] but notably lower compared to China (57%) [26]. In younger bovines, higher prevalence such as 58.5% in calves [6] and 39% in heifers [30] have been reported in the USA.

*C. pecorum* was first detected in a cow with neurological symptoms due to the so-called "Buss-Encephalitis", which was later renamed as sporadic bovine encephalomyelitis [31]. Other clinical signs in cattle are polyarthritis, enteritis, keratoconjunctivitis, mastitis, respiratory signs, and fertility problems because of endometritis or abortions [12, 18, 32]. Altogether, the virulence of *C. pecorum* is low leading to multifactorial diseases in which co-infections, management, hygiene, and stress are important factors determining the presence and severity of clinical signs [25].

For diagnostic purposes, direct pathogen detection by molecular methods is the method of choice [33]. A swab sample is suitable to collect samples from rectal, vaginal, or conjunctival sites. During sampling, it is important to ensure that epithelial cells are obtained, as the bacteria mainly reside intracellularly. Subsequently, the swab should be tested by either a screening *Chlamydiaceae*-PCR combined with either microarray, single- or multi-locus sequencing, or by a species-specific qPCR to identify *C. pecorum* on species-level. In contrast to serological or cultural detection methods, PCR methods are faster, more sensitive, and more specific [33, 34].

Limited data are available on the worldwide prevalence of *C. pecorum* in cattle, particularly in European countries. Moreover, the presence of *C. pecorum* has not been investigated in Switzerland to date. Therefore, the aim of this study was to gain an insight into the *C. pecorum* prevalence in a defined but not preselected healthy Swiss cattle population. Furthermore, we wanted to investigate the shedding dynamics and selected influencing factors as part of our longitudinal study. This was complemented by examinations of the shedding route (conjunctival versus rectal) in different age groups as well as the assessment of bacterial loads.

## Material and methods

### Study design, farm characteristics and animals

During this longitudinal study, the bovine herd located at the AgroVet-Strickhof in Lindau, ZH, in the north-eastern part of Switzerland, was repeatedly sampled over a one-year period in three-month intervals (Timepoint 1–5, T1-5). At all timepoints, the bovines on the farm were tested by collecting a conjunctival as well as a rectal swab. The study was approved by the cantonal veterinary office Zurich (TVB Nr. ZH223/2020).

The AgroVet-Strickhof contains a bovine population of approximately 140 dairy cows, beef cattle and heifers, with annual new stockings of about 240 calves. In contrast to the average dairy farm size in Switzerland of about 30.9 dairy cows per farm (actualized at 2022) [35], the AgroVet-Strickhof represents a bigger farmyard. On the farm, other animals such as horses, sheep, poultry, pigs, and alpacas are held but without direct contact to the bovine population.

**Dairy cows.** The population of the dairy cows was defined as all bovines being part of the dairy herd. All cows in this category calved at least once except for eight heifers which were integrated before calving during the third and fifth sampling. The age ranged from less than two years old bovines which were right before calving to almost 14 years old cows. At the last sampling timepoint (T5), the average age was approximately four and a half years. The dairy cows were kept in a free-range housing system with around half of the herd having seasonal access to separated grassland. The herd consisted of different breeds, but the Original Swiss Browns, Brown Swiss, Holstein, and Red Holstein breeds predominated. The average milk yield of the farm is 10,029 L/cow/year (actualized at 2022) [36]. The range of the number of lactations reached from one to ten. For hygienic purposes, isolation and calving pens with deep litter were used. The feeding consisted of a total mixed ration (TMR) in addition to daily hay. The TMR consisted of corn and grass silage, concentrated feed, alfalfa, straw as well as mineral feed. The total number of animals in the dairy herd remained relatively stable ranging between 88 and 135. New additions were self-reared, and no dairy cows were purchased. The most common reasons for culling were common health problems such as infertility as well as udder and claw diseases on an individual basis. Otherwise, the population was clinically healthy.

**Beef cattle.** The beef cattle group was the most variable in this study. Their age ranged from 108 days (d) to almost two years (725 d). This category contained all animals which were kept for meat production. At T1, a total of 26 heifers were added to the category beef cattle in addition because of their age range and the proximity to the remaining beef cattle herd. During this study, 16 of these heifers were integrated into the dairy herd. Beef cattle were mostly purchased from different farms all over Switzerland and only few of them were self-reared. Most of the beef cattle were crossbreeds containing Simmental. The beef cattle population was kept in an indoor housing system. The feeding comprised an externally produced TMR containing concentrated feed, straw, corn, and grass silage. In addition, hay was offered *ad libitum*. The beef cattle were sent to slaughter at a body weight of at least 530 kg. Simmental crossbreed slaughter weight was slightly higher with 560 kg.

**Calves.** The calves were housed in groups of five to ten animals and kept on straw bedding. The age ranged from 36 days to 169 days. Calves younger than 36 days could not be sampled because they were kept separately in igloos for the first 4–5 weeks for hygienic reasons. In this category, male calves were mostly sold to fatteners and only a few were transferred to the company's own fattening unit. The female calves were mostly sold to nursery farms and later returned to the dairy herd at AgroVet-Strickhof. Only a few calves were kept on the farm and raised in-house. During our study, nine calves were integrated into the beef cattle population.

In total, the study included five sampling timepoints between 09.04.2021 and 17.05.2022 covering a full year. A total of 308 different bovines were tested during this time period, with 202 of them tested at least twice, resulting in 782 samples collected from both sampled areas (S1 Table). In particular, the dairy cows were sampled several times as they formed a relatively stable group in contrast to younger bovines on this farm. The population of the dairy herd was also the biggest group with an average of 111 animals. The average beef cattle group size per sampling timepoint was 34, while the calf population consisted of 12 animals on average. Detailed animal numbers at each sampling timepoint are shown in S2 Table.

Each animal was sampled with one rectal and one conjunctival swab. To obtain the desired rectal sample material, the swab was inserted approximately 5 cm and gently pressed against the dorsal part of the rectum where it was rotated for about three seconds. During sampling, a second swab was used for the conjunctival sample. This swab was inserted in the median corner of one eye and rotated under gentle pressure for about three seconds. The same swab was then used to sample the other eye of the same animal. The collected samples were then labelled with the date, animal number and rectum/conjunctiva sample (R / C), placed on ice and brought to the laboratory on the same day. The samples were then transferred to -20˚C until further processing. All bovines/sampling sites were tested with FLOQSwabs Cat. No. 961C (COPAN ITALIA spa, Brescia) except for the conjunctival swabs of calves, which were collected with a smaller swab by the same company (COPAN ITALIA spa; Cat. No. 551C).

## DNA extraction

For DNA extraction, the Maxwell® 16 buccal swab LEV DNA purification kit, #AS1295 (Promega, Madison, WI, USA) was used according to manufacturer's instructions. First, a clearing column was added to a 1.5 ml Maxwell microtube. Each sample was then put into a clearing column/microtube container by clipping the swab tip with a scissor. To reduce DNA contamination, the instruments and the working area were washed with $H_2O$, 70% Ethanol and DNA-away between every sample or every run, respectively. In a separate tube, 300 μl lysis buffer and 30 μl proteinase K was prepared for each sample. A total of 330 μl was then added to the sample which was subsequently vortexed for 10 s and incubated in a ThermoMixer™ at 56˚C for 20 min. Next, DNA was extracted using the Maxwell instrument and eluted in 50 μl elution buffer.

## Nanodrop-1000 & dilution

DNA concentration and quality were determined using the Nanodrop® 1000 Version 3.7.1 (Thermo Fisher Scientific; Waltham, MA, USA). Before each measurement, the machine was calibrated with Elution buffer of the DNA extraction. All rectal swab samples and all conjunctival swab samples showing concentrations > 150 ng/μl were diluted with molecular grade $H_2O$ for later qPCR (Table 1). After dilution the eluate was stored at -20˚C until further use.

## *Chlamydiaceae*-specific 23S rRNA qPCR

All swab samples taken at the first two sampling timepoints (n = 654) were assessed by the *Chlamydiaceae* family-specific 23S qPCR for screening purposes. This qPCR was performed

**Table 1. Dilutions used for PCR according to the quantity of the extracted DNA determined by nanodrop.**

| Sampling site | 1:100 | 1:10 | 1 (not diluted) |
|---|---|---|---|
| Rectum | >120 ng/μl | <120 ng/μl | - |
| Conjunctiva | - | >150 ng/μl | <150 ng/μl |

on an ABI 7500 instrument (Applied Biosystems, Forster City, CA, USA) according to the protocol of Ehricht et al. [37], targeting a 111 bp sequence of the 23S ribosomal RNA (rRNA). Additionally, an internal amplification control was added to each qPCR run [38]. The primers and probes are listed in S3 Table. Every sample was tested in duplicate with the cycle threshold value set at 0.1 for each test run. The cutoff for positive samples was set at a mean cycle threshold (Ct-value) of 38. For quantification purposes, a seven-fold *C. abortus* standard curve was used and $H_2O$ served as negative control. If the amplification control showed signs of inhibition, the qPCR was repeated at a higher dilution as in Table 1.

### *C. pecorum*-specific qPCR

The species-specific *C. pecorum* qPCR targets a 76 bp sequence within the *ompA* gene [3] and was performed on each sample collected during the five sampling timepoints (n = 1,564). The ABI 7500 was also used for this qPCR which was performed according to Pantchev et al. [3]. Based on evaluations to determine the limit of detection (LOD) shown in S4 Table, the Ct-value cutoff was set at 38. The internal amplification control and the qPCR protocol were performed as previously described [39]. Standard curves were generated using a sevenfold dilution series of *C. pecorum* recombinant plasmid DNA as described in Rohner et al. [39]. $H_2O$ was again used as negative control. All primers and probes used are listed in S3 Table.

### Species identification by 16S rRNA conventional PCR and sequencing

During the first two sampling timepoints, all samples which were tested positive for *Chlamydiaceae* but negative for *C. pecorum* (n = 1) were further examined by the 16S rRNA conventional PCR as described by Taylor-Brown et al. [40]. Subsequently, Sanger sequencing of the PCR products was performed by Microsynth AG (Balgach, Switzerland) and processed as described by Rohner et al. [39] to further determine the *Chlamydia* species. Consensus sequences were generated by *de novo* assembly using Geneious Prime (Version 2021.1; Biomatters, Auckland, New Zealand). Sequences were then compared to known sequences in the NCBI database by BLASTn analysis [41].

### Quantification of *C. pecorum* copy number per µl and per swab

The average copy number per samples was determined as described in Rohner et al. [39]. Briefly, we determined the y-intercept and exponential growth constant using the standard curve of each run. The average number of copies (noc) per µl as well as per swab (50 µl elution tube) was then calculated using the average Ct-value per sample. All determined loads were further divided into three categories based on their shedding amount. The low shedders were defined as all samples with <100 noc/µl, and moderate shedders were all samples between 100–1,000 noc/µl. All samples with >1,000 noc/µl were classified as high shedders as reported previously [42].

### Statistics

Statistics was performed using the Stata® program (StataCorp., 2021; Stata Statistical Software®: Release 17.0; College Station, TX, USA: StataCorp MP). First, a quality control of the data and the descriptive analysis was done using <codebook varx₁ varxₙ>, where varx₁ to varxₙ represent the variables from $x_1$ to $x_n$. Univariate analysis for categorical data was performed using chi-square or Fisher's exact (for cells n < 5) test and for continuous data t-test or ANOVA (analysis for variance, <oneway>) with Bonferroni post hoc correction. Linear regression was also performed. A p-value of ≤ .05 was considered as significant. A power of > .8 was considered.

## Results

### *C. pecorum* was the only chlamydial species found in the bovine AgroVet-Strickhof population

To identify the different chlamydial species present at AgroVet-Strickhof, all swabs taken during the first two sampling timepoints (n = 654) were tested with a *Chlamydiaceae*-specific qPCR, resulting in 15.1% positive swabs (99/654). Subsequently, a *C. pecorum* species-specific qPCR was performed on all samples, in which 100/654 (15.3%) samples were positive. With three exceptions, the samples were always positive for both family-specific and *C. pecorum*-specific qPCR or negative in both tests performed. The exceptions included a rectal sample (beef cattle no. 1471) and a conjunctival sample (beef cattle no. 1487) at T1. These two samples tested negative for *Chlamydiaceae* but showed a *C. pecorum*-positive result. The mean Ct-values in the *C. pecorum* qPCR were 35.98 and 36.11 for sample no. 1471 and 1487, respectively. In the 23S qPCR, no. 1471 could not be detected, whereas no. 1487 had a 23S Ct-value of 38.46, which was just above the threshold value of 38 and was therefore classified as negative. Another discrepancy was detected in the rectum sample from dairy cow no. 351 at T1. This sample tested positive for *Chlamydiaceae* with a mean Ct-value of 35.85 but was negative using the *C. pecorum* qPCR. To further determine the chlamydial species present, a conventional 16S rRNA PCR and subsequent sequencing was performed on this sample. However, no relevant sequence was detected in the sequencing and the species could not be determined, likely due to low chlamydial loads.

### The highest *C. pecorum* prevalence was found in calves, whereas dairy cows had the lowest prevalence

During this study, rectal and conjunctival swabs were taken from 782 animals, resulting in a total of 1,564 samples. Overall, 216 (13.8%) samples tested positive for *C. pecorum*. Prevalences were determined as animal prevalences (Fig 1A), as well as rectal (Fig 1B) and conjunctival prevalence (Fig 1C), based on the corresponding sampling site. For the animal prevalence, an animal was considered positive if at least one anatomical location (rectal and/or conjunctival) was positive by qPCR.

In dairy cows, we detected no positive rectal samples resulting in an animal prevalence (Fig 1A) that was only determined by the conjunctival prevalence; it ranged between 5.2 and 11.4%. Animal prevalences in beef cattle ranged between 38.1 and 61.5%. The animal prevalence in calves was generally high, except for T4 (55.0%), and even reached 100% at three sampling timepoints. In contrast to dairy cows without any rectal positivity, beef cattle mostly had a rectal prevalence ranging between 10 and 20%, while the rectal prevalences in calves usually exceeded 60% (Fig 1B). The only exception was at T4 where beef cattle had a rectal prevalence of 47.1% and therefore higher than the prevalence found in calves (25%). The conjunctival prevalence of dairy cows remained stable with values ranging between 5.2 and 11.4% during all sampling timepoints. In the younger age categories, the conjunctival prevalence was 23.5–59% and $\geq$50% in beef cattle and calves, respectively (Fig 1C). Generally, rectal prevalences were lower than the ocular prevalences in all age categories. Furthermore, rectal prevalence showed less fluctuations compared to conjunctival prevalence. Next, the differences between the age categories were statistically evaluated by the Fisher's exact or Pearson Chi$^2$ test depending on group sizes. The prevalence differences between dairy cows and beef cattle, and between dairy cows and calves, were statistically significant except for the conjunctival prevalence compared between dairy cows and beef cattle at T4 (p = 0.059). A comparison between the age categories of beef cattle and calves showed a significantly higher prevalence in calves except for

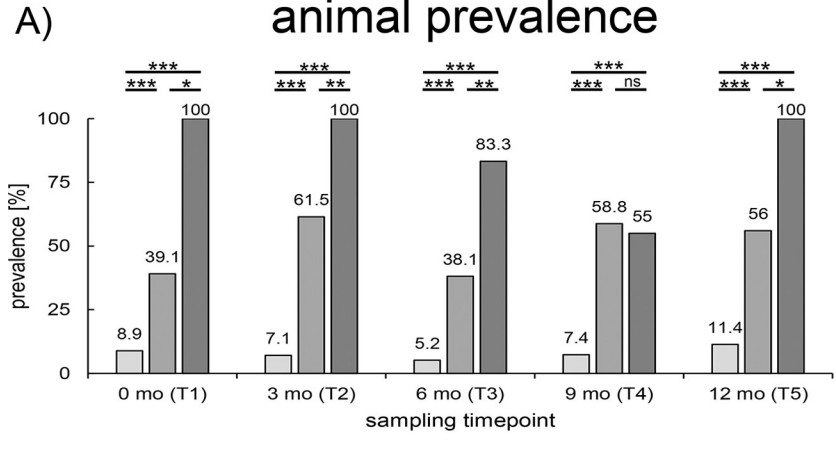

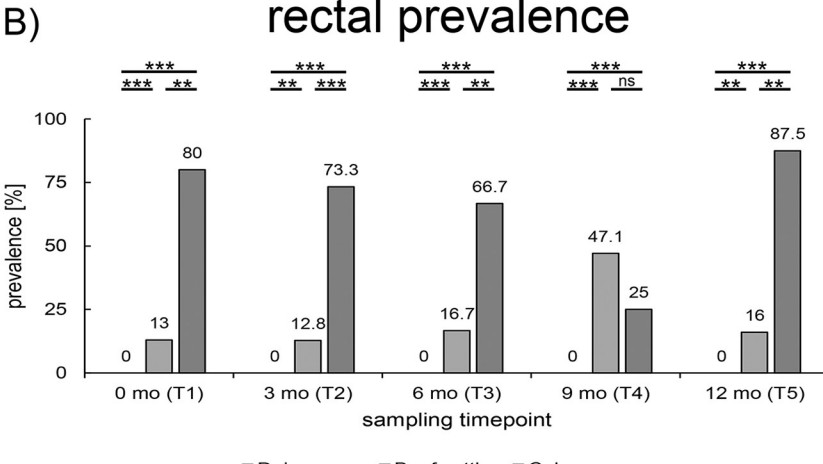

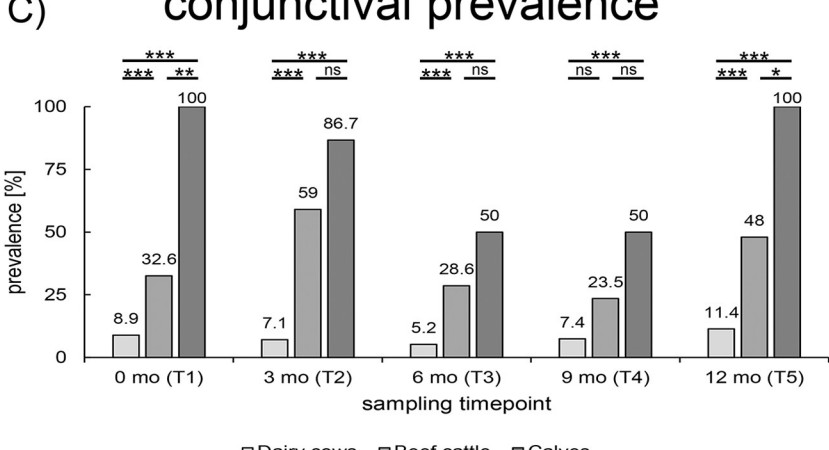

**Fig 1. *C. pecorum* prevalence.** Shown are the different *C. pecorum* prevalences in each age category at all sampling timepoints. Prevalences are displayed as (A) animal prevalence, (B) rectal prevalence and (C) conjunctival prevalence. For animal prevalence, an animal was defined as positive if at least one sample (rectal and/or conjunctival) was positive. Significant differences were represented with asterisks: Three asterisks (***) represent p-values <0.001, two asterisks (**) represent p-values <0.01 and one (*) represents p-values between 0.01 and 0.05. Non-significant values were labeled with ns.

timepoint T4 in animal prevalence (p = 1) as well as in rectal prevalence (p = 0.161). Significantly higher conjunctival prevalences could only be shown in calves at T1 and T5 compared to beef cattle. All corresponding p-values of the comparisons are listed in S5 Table. In summary, the youngest age category (calves) showed the highest *C. pecorum* prevalence followed by beef cattle and dairy cows.

## *Chlamydia pecorum* can be detected at a higher prevalence in conjunctival than in rectal samples

A total of 65 rectal and 151 conjunctival swabs were positive for *C. pecorum* at the five sampling timepoints (Fig 2). In beef cattle and calves, *C. pecorum* was always found at both anatomical localizations (Table 2) whereas *C. pecorum* was only detected in conjunctival swabs of dairy cows.

To compare the distribution of the positive samples related to their sampled location, Fisher's exact and Pearson Chi$^2$ test were carried out. Considering all age categories, the amount of positive conjunctival swabs was significantly higher compared to rectal swabs at all sampling timepoints (Fig 2). The calculated p-values were p < 0.001 at T1, T2, T3 and T5 and p = 0.001 at T4. In addition, the distribution within each age category was assessed statistically except for the category of dairy cows (no rectal positive swabs). Similarly, no statistics could be performed within the calf category at T1 and T5 because they had 100% positivity in conjunctival swabs. At the other sampling timepoints, there was no significant difference between rectal and conjunctival samples in calves (p = 1 at T2 and T3 and p = 0.303 at T4). Similar results were observed in beef cattle, where no difference regarding the anatomical localization was observed (p = 0.375 (T1), p = 0.631 (T2), p = 0.387 (T3) and p = 1 (T4, T5)).

## Influencing factors on the *C. pecorum* positivity rate

To gain more knowledge about *C. pecorum* infection dynamics, the following factors were investigated in more detail: age and breed of the sampled animals, weight, and acquisition

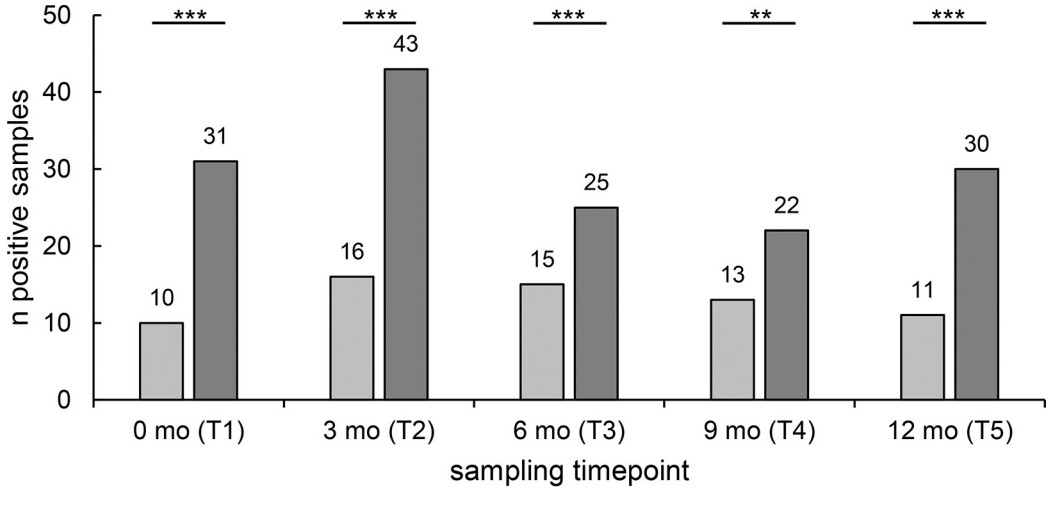

**Fig 2. Total number of *C. pecorum*-positive samples at the rectal and conjunctival sampling site.** The number of *C. pecorum*-positive samples for both anatomical sites at all five sampling timepoints are shown. In total, 65 positive samples were taken from the rectum, and 151 positive samples were taken from the conjunctiva. Significant differences were represented with asterisks: Three asterisks (***) represent p-values <0.001, two asterisks (**) represent p-values <0.01 and one (*) represents p-values between 0.01 and 0.05. Non-significant values were labeled with ns.

**Table 2. Number of all positive samples divided by age category and anatomical location.**

| Category | Swab | T1 | T2 | T3 | T4 | T5 | Total |
|---|---|---|---|---|---|---|---|
| Dairy cows | R | 0/123 | 0/99 | 0/135 | 0/108 | 0/88 | 0/ 553 |
|  |  | (0%) | (0%) | (0%) | (0%) | (0%) | (0%) |
|  | C | 11/123 | 7/99 | 7/135 | 8/108 | 10/88 | 43/553 |
|  |  | (9%) | (7%) | (5%) | (9%) | (11%) | (8%) |
| Beef cattle | R | 6/46 | 5/39 | 7/42 | 8/17 | 4/25 | 30/169 |
|  |  | (13%) | (13%) | (17%) | (47%) | (16%) | (18%) |
|  | C | 15/46 | 23/39 | 12/42 | 4/17 | 12/25 | 66/169 |
|  |  | (33%) | (59%) | (29%) | (24%) | (48%) | (39%) |
| Calves | R | 4/5 | 11/15 | 8/12 | 5/20 | 7/8 | 35/60 |
|  |  | (80%) | (73%) | (67%) | (25%) | (88%) | (58%) |
|  | C | 5/5 | 13/15 | 6/12 | 10/20 | 8/8 | 42/60 |
|  |  | 100%) | (87%) | (50%) | (50%) | (100%) | (70%) |
| Total |  | 41/348 | 59/306 | 40/378 | 35/290 | 41/242 | 216/1564 |

R = rectal swab, C = conjunctival swab, T1-5 = sampling timepoint T1 (0 months), T2 (3 months), T3 (6 months), T4 (9 months), T5 (12 months)

status (self-reared vs. purchased) of beef cattle as well as lactation status and lactation number of dairy cows.

**The probability of being *C. pecorum*-positive correlated with age.** First, a t-test was applied to test for a correlation between age and the probability of a positive *C. pecorum* qPCR result. During all sampling timepoints, the age ranged between 36 d (Calves no. 1946 and 1947 at T4) and 4939 d (Dairy cow no. 3 at T5). Across all categories, there was a significant correlation between age and positivity for animal (Fig 3A) and sample prevalence at all sampling timepoints (p-values < 0.001 as shown in S6 Table). As an example, according to animal prevalence at T1, the positive population (n = 34) showed an average age of 651 d, whereas the negative population (n = 140) was on average 1502 days and thus more than twice as old. Within individual age categories, a significant correlation between younger age and a higher *C. pecorum* prevalence (animal prevalence) was only found in the beef cattle category at T1 (p = 0.012) and T3 (p < 0.001). No significant correlations were found in the age categories of dairy cows or calves. However, it must be noted that the examination of calves could not be carried out at T1, T2 and T5, as the animal prevalence was 100% in each case and a negative group was missing. All corresponding p-values are listed in S7 Table.

**The probability of being *C. pecorum*-positive correlated with body weight in beef cattle only at T3.** Excluding the 26 heifers added at T1, the weights of all beef cattle were recorded and ranged between 150 kg (no. 5422 at T1) and 570 kg (no. 2882 at T4). As shown in Fig 3B, *C. pecorum*-positive animals were always found to be lighter on average compared to negative beef cattle. However, statistical analysis with a oneway ANOVA only showed a significant correlation between animal prevalence and weight at T3 (p < 0.001). At the other sampling timepoints, the differences at T1, T2, T4 and T5 were not statistically significant (p-values of 0.108, 0.688, 0.863 and 0.098).

**Own-breed beef cattle were more often positive than purchased ones.** Next, the correlation between the *C. pecorum* prevalence and the status of the purchase was examined. Since all animals in the groups of dairy cows and calves were self-reared, this evaluation could only be carried out in the beef cattle category. A total of 169 samples per localization were taken in this category. Of these, 104 of the samples were taken from animals that were purchased, while the remaining 65 samples were taken from self-reared beef cattle. Apart from T1, beef cattle were

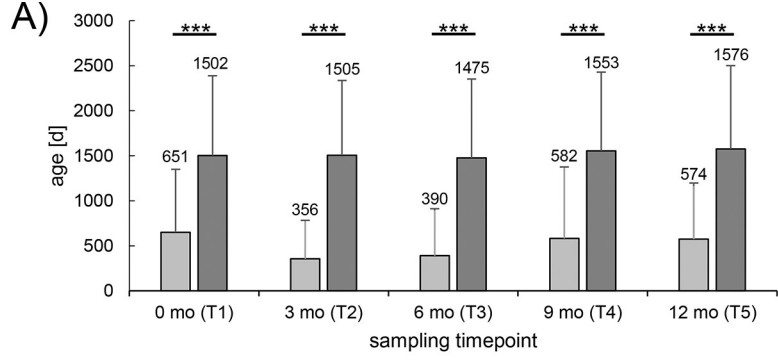

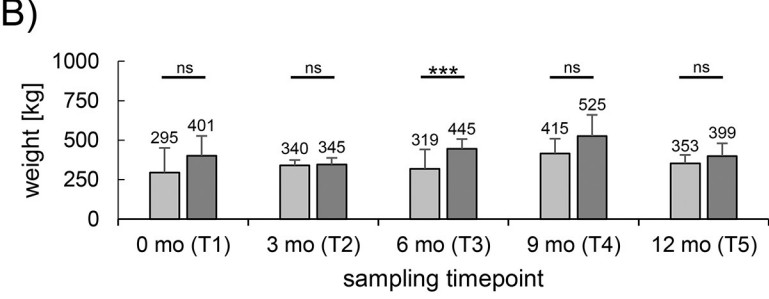

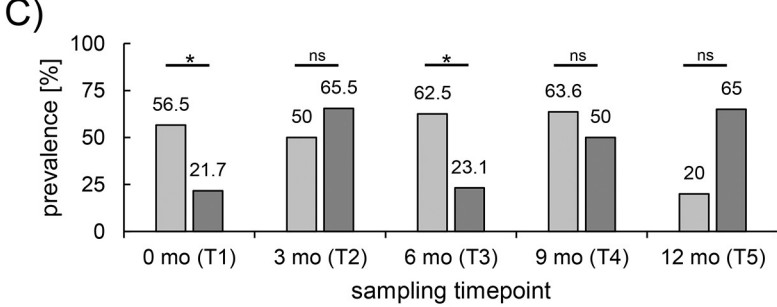

**Fig 3. Influencing factors on *C. pecorum* prevalence.** Shown is the comparison of the (A) age and (B) weight between *C. pecorum*-negative and *C. pecorum*-positive bovines investigated in this study (mean ± standard deviation). (C) The graph compares the prevalence between self-reared and purchased beef cattle during the five sampling timepoints. Significant differences were represented with asterisks: Three asterisks (***) represent p-values <0.001, two asterisks (**) represent p-values <0.01 and one (*) represents p-values between 0.01 and 0.05. Non-significant values were labeled with ns.

purchased in small groups or as individual animals from different farms. Only at T1, larger groups were purchased from the same farm (farm of origin 1, n = 13; farm of origin 2, n = 10). The following numbers of beef cattle were purchased at the different sampling times: T1 (23/46), T2 (29/39), T3 (26/42), T4 (6/17) and T5 (20/25).

As shown in Fig 3C, the *C. pecorum* prevalence was distributed differently depending on the sampling timepoint. A statistical analysis using a Fisher's exact or Pearson Chi$^2$ test showed that only the differences at T1 (p = 0.033) and T3 (p = 0.011) were significant, whereas other timepoints showed no correlation (T2 (p = 0.463), T4 (p = 0.644), T5 (p = 0.133)). At both

sampling timepoints, the self-reared beef cattle had a higher animal prevalence than purchased cattle. At T1, 13/23 self-reared beef cattle were positive, whereas only 5/23 of the purchased cattle were positive. At T3, 10/16 self-reared and 6/26 purchased beef cattle were positive.

**Other conditions like breed, lactation status and lactation number did not show any significant correlation within the dairy herd.** Next, a correlation between prevalence and breed was tested in all categories by using Fisher's exact or Pearson Chi$^2$ test, but no significant correlation with the animal prevalence was found at any timepoint. Investigating dairy cows, we found no correlation between the *C. pecorum* positivity and lactation status or number at any sampling timepoint. The corresponding p-values for the different comparisons are listed in S8 Table.

## Chlamydial loads

**Rectal and conjunctival chlamydial loads were higher in calves than in other age categories.** We then determined the chlamydial load expressed as *C. pecorum*-copies per μl and compared average loads between age categories. Therefore, the loads were considered as animal-independent variables and evaluated collectively at all sampling timepoints. The calculations were carried out with absolute noc as well as the mean values of the different sampling timepoints in order to identify potential outliers.

Overall timepoints, the mean rectal load was 296.2 noc/μl and 2,743.7 noc/μl for beef cattle and calves, respectively. The mean conjunctival loads including all sampling timepoints were 672.8 noc/μl, 323.6 noc/μl and 1,739.7 noc/μl in dairy cows, beef cattle and calves, respectively (Fig 4A and 4B). A detailed list of the mean loads and the corresponding standard deviations at all different sampling timepoints is shown in S1 Fig and S9 Table. The highest rectal loads were observed in calves at every sampling timepoint. The highest conjunctival loads were also found in calves except for T3, where dairy cows had a higher load. The lowest conjunctival loads were observed in beef cattle, except for T5. However, including all sampling timepoints there was a significantly higher noc in the rectal samples of calves (oneway ANOVA), regardless of whether absolute (p = 0.033, S2A Fig) or mean loads (p < 0.001, Fig 4A) were used for statistical analysis. In contrast, comparing the absolute conjunctival *C. pecorum* loads between the different age categories showed no significance. Specifically, p-values included 0.166, 0.563 and 1 for the comparison of calves with beef cattle, calves with dairy cows and beef cattle with dairy cows, respectively (S2B Fig). However, when the comparison for conjunctival loads was performed based on mean values, calves had a significantly higher load than the other age categories (each p < 0.001, Fig 4B). The difference between dairy cows and beef cattle was not significant with p = 0.203 (Fig 4B).

**Younger animals had higher mean loads than older animals.** To test for a correlation between age and loads, a regression analysis (raw data in S1 Text) was performed across all animals. When calculating with absolute loads, the rectal loads showed a tendency to be higher in younger animals with p = 0.056 (S3 Fig). In contrast, the conjunctival samples showed no significant correlation between the load and the age with p = 0.491 (S3 Fig). However, when using the mean loads, both the rectal samples (p < 0.001) and the ocular samples (p = 0.028) showed a significantly higher load in younger animals. Furthermore, comparing the loads and the age within the individual categories, the only significant value (p = 0.013) was found in calves where the mean conjunctival load correlated with age. All corresponding p-values are listed in S10 Table.

**Mean loads were significantly higher at the rectal compared to the conjunctival sampling site.** The loads of the rectal and conjunctival sampling sites were compared across all animals and within age categories using an unpaired t-test (Fig 5). If the calculations were

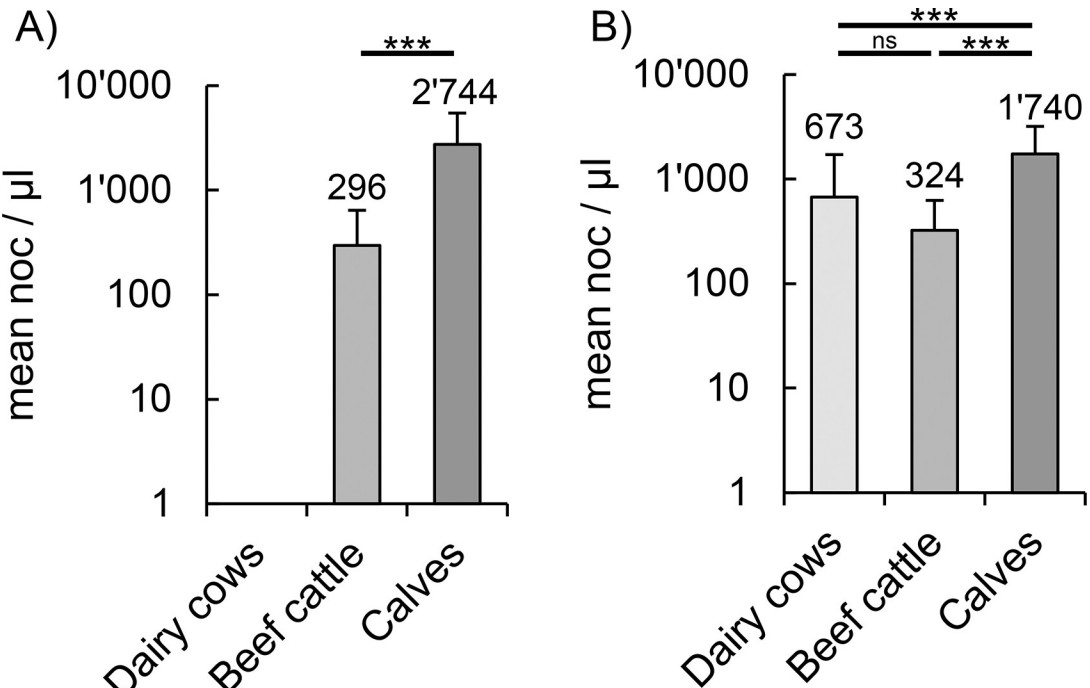

**Fig 4. Mean *C. pecorum* loads.** Shown is the (A) rectal and (B) conjunctival *C. pecorum* load as number of copies (noc) per μl for each age category including all sampling timepoints (logarithmic scale, mean ± standard deviation). Statistical analyses were performed using the mean loads of each respective sampling timepoint. Significant differences were represented with asterisks: Three asterisks (***) represent p-values <0.001, two asterisks (**) represent p-values <0.01 and one (*) represents p-values between 0.01 and 0.05. Non-significant values were labeled with ns.

done using absolute loads, no significant difference between the rectal and conjunctival loads was seen in any age category comparison (Fig 5). To adjust for the large variation of the loads, the statistics were also carried out using the mean values. This approach showed a statistically higher rectal load within calves (p = 0.043) and across all age categories (p < 0.001). Mean values containing all categories were 1,614.1 noc/μl in rectal and 816.9 noc/μl in conjunctival samples. In calves, mean values were 2,743.7 noc/μl in rectal and 1,739.7 noc/μl in conjunctival swab samples. All corresponding p-values are listed in S11 Table.

**Subdivision in low, moderate, and high shedders.** Finally, the 216 positive samples were divided into three shedding categories as performed in Bommana et al. [43]: <100 noc/μl (C1, low shedder), 100–1,000 noc/μl (C2, moderate) and >1,000 noc/μl (C3, high). Our obtained range of loads was 4.7–34,531.9 noc/μl at the rectal and 0.8–37,743 noc/μl at the conjunctival site. Out of a total of 216 samples, 144 were categorized as C1, while both C2 and C3 included 36 samples (Table 3). The 36 samples in the high-shedding category (C3), corresponded to 16.7% of all *C. pecorum*-positive and 2.3% of all samples taken, and included 16 from the rectal and 20 from the conjunctival sampling site. However, considering that seven calves were high shedders at both locations during one sampling timepoint, only 9.4% (29/308) of all individually sampled bovines belonged to C3. Twenty of these high shedders were calves, five were beef cattle and four belonged to the dairy cows. Like C3, C2 also had a total of 36 samples. Regarding the localization, they were equally distributed at both anatomical sites (n = 18 each). The biggest category by far was the low-shedding category C1 with a total of 144 positive samples. C1 contained 31 rectal and 113 conjunctival samples which equals nearly 50% and 75% of all positive samples at the corresponding anatomical site.

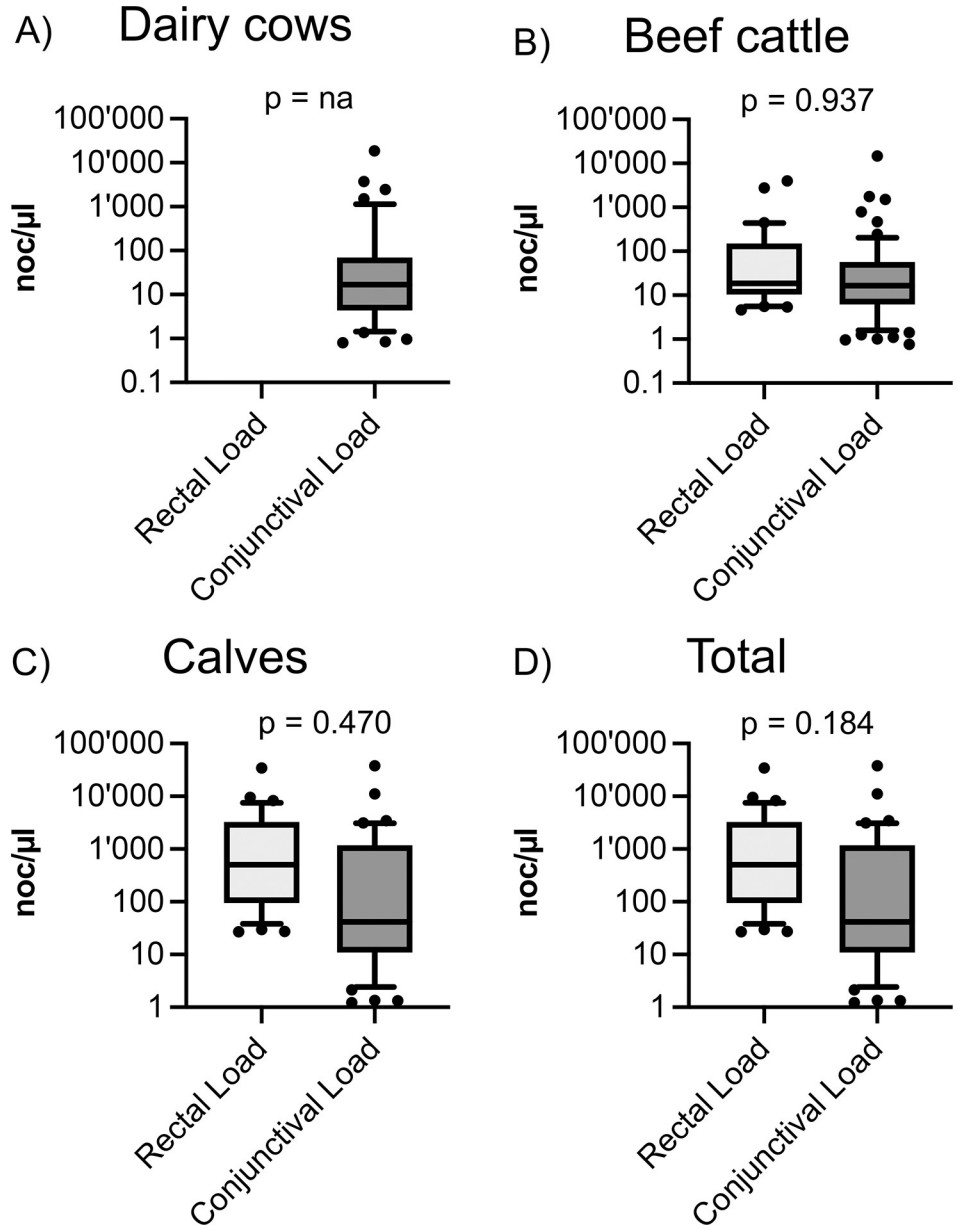

**Fig 5. *C. pecorum* loads based on anatomical localization.** Shown are the number of copies (noc) per µl of the different localizations (rectal, conjunctival), both within (A) dairy cows, (B) beef cattle, (C) calves and (D) across all animals. The p-values were obtained by comparing the absolute loads and showed no significant difference.

Furthermore, we investigated the shedding category distribution within dairy cows, beef cattle and calves depending on the sampling site. In dairy cows, *C. pecorum* positive samples were only detected in conjunctival swabs and were mostly detected in C1 (33/43) and only few samples belonged to C2 and C3 with six and four samples, respectively (Table 3). In beef cattle, we found a similar pattern for both anatomical sites with >70% of samples belonging to C1, 10–20% to C2 and <10% to C3. In contrast, most rectal samples in calves were categorized as C3 (14/35) followed by C2 (12/35) and C1 (9/35). For conjunctival samples, calves revealed an unusual biphasic pattern where most samples belonged to C1 (24/42) followed by C3 (13/42) and C2 (5/42).

**Table 3. Classification into low, moderate, and high shedders.**

| Category | N positive samples (R/C) | Low noc <100 | | Moderate noc 100–1,000 | | High noc >1,000 | |
|---|---|---|---|---|---|---|---|
| | | R | C | R | C | R | C |
| Dairy cows | 43 (0/43) | -[1] | 33 | - | 6 | - | 4 |
| | | | (76.7%)[2] | | (14%) | | (9.3%) |
| Beef cattle | 96 (30/66) | 22 | 56 | 6 | 7 | 2 | 3 |
| | | (73.3%) | (84.8%) | (20%) | (10.6%) | (6.7%) | (4.5%) |
| Calves | 77 (35/42) | 9 | 24 | 12 | 5 | 14 | 13 |
| | | (25.7%) | (57.1%) | (34.3%) | (11.9%) | (40%) | (31%) |
| Total | 216 (65/151) | 31 | 113 | 18 | 18 | 16 | 20 |
| | | (47.7%) | (74.8%) | (27.7%) | (11.9%) | (24.6%) | (13.2%) |

[1] All samples were negative.

[2] Absolute sample numbers as well as percentages are shown.

## Animals at all age groups were positive at multiple timepoints

During this study, 134 bovines were tested positive for *C. pecorum* at least once. Most of them (n = 103) tested positive only at one sampling timepoint, whereas 31 bovines were positive multiple times. Out of these 31 bovines six were dairy cows (Table 4), 14 were beef cattle (Table 5) and eleven were calves (Table 6). In dairy cows, six animals were positive at multiple timepoints, five of them twice. All positive samples (n = 14) were taken from the conjunctiva and *C. pecorum* loads ranged from 0.8 to 272.6 noc/μl. The animal no. 386 was positive four times over a period of nine months (T2-T5), but always with low loads. The loads in dairy cows positive at multiple timepoints were generally low (Table 4). Only three animals (no. 309, 316 at T5 and 377 at T1) had values >100 noc/μl. In animal number 377, a high conjunctival load at T1 (272.6 noc/μl) was measured followed by a drastic decrease to 2.5 noc/μl within three months and was then negative at the following sampling timepoints. For animal number 309 and 316, a high load was observed at the last sampling timepoint.

In the age category of beef cattle (defined by category during the first sampling), a total of 14 animals tested positive several times (Table 5). Of these, 13 animals were positive twice and one animal (no. 5421) was positive three times. Out of 31 positive samples, eight were taken

**Table 4. Dairy cows which were positive multiple times.**

| Loads per μl in dairy cows (R/C) | | | | | |
|---|---|---|---|---|---|
| Animal ID | T1 | T2 | T3 | T4 | T5 |
| 309 | -/-[1] | -/16.8[2] | -/- | -/- | -/236.1 |
| 316 | -/0.8 | -/- | -/- | -/- | -/125.6 |
| 377 | -/272.6 | -/2.5 | -/- | na[3] | -/- |
| 383 | -/7.3 | -/- | -/69.2 | -/- | na |
| 386 | na | -/7.3 | -/13.7 | -/4.8 | -/4.3 |
| 397 | na | na | -/56.8 | -/14.9 | na |

[1] Both samples showed a negative *C. pecorum* qPCR result.

[2] Gray shading indicates a positive result in rectal and/or conjunctival (R/C) samples with corresponding *C. pecorum* load per μl.

[3] Animal was not available.

**Table 5. Beef cattle which were positive multiple times.**

| Loads per μl in beef cattle (R/C) | | | | | |
|---|---|---|---|---|---|
| Animal ID | T1 | T2 | T3 | T4 | T5 |
| 1431* | -/12.0[1] | na[2] | na | -/22.2 | na |
| 1436* | -/-[3] | na | -/20.7 | -/- | -/57.4 |
| 1626 | -/8.2 | -/22.4 | na | na | na |
| 2734 | Na | -/- | 183.6/- | 47.4/- | na |
| 3626 | Na | -/4.4 | -/- | -/8.3 | na |
| 4898 | Na | -/- | -/48.1 | 395.6/- | na |
| 5172 | Na | -/37.3 | -/67.4 | na | na |
| 5421 | -/5.0 | -/6.2 | -/34.9 | na | na |
| 5422 | 39.2/- | -/- | -/- | 5.4/- | na |
| 5423 | -/57.1 | -/13.5 | -/- | na | na |
| 5461 | Na | na | 3,974.6/23.2 | -/- | -/15.3 |
| 5824 | Na | 12.1/244.1 | -/10.9 | -/- | na |
| 6768 | Na | -/57.3 | 9.5/- | na | na |
| 8753 | Na | -/187.4 | -/467.6 | na | na |

[1] Gray shading indicates a positive result in rectal and/or conjunctival (R/C) samples with corresponding *C. pecorum* load per μl.

[2] Animal was not available.

[3] Both samples showed a negative *C. pecorum* qPCR result.

* Beef cattle that were integrated into the dairy cow population during this study.

rectally and 23 from the conjunctiva. Two animals were tested positive at both locations, whereas six were only positive rectally and 21 were only positive at the conjunctival site. Rectal loads ranged from 5.4 to 3,974.6 noc/μl, with three values >100 noc/μl. The highest rectal load was measured in animal no. 5461 at T3 and was negative in the subsequent sampling three months later. The conjunctival loads ranged from 4.4 to 467.6 noc/μl, with only three samples having a load >100 noc/μl. Beef cattle no. 8753 was the only animal that showed a load of >100 noc/μl on two consecutive samples (T2 and T3).

Among the eleven calves that tested positive multiple times, five were positive twice and six were positive three times (Table 6). Nine of them joined the population at T2 and two calves joined at T3. All calves tested positive for at least two consecutive samplings. After that, four calves showed a negative result, whereas the others left the population and were not sampled again. The resulting 40 positive samples were relatively evenly distributed regarding their sampling localization with 19 rectal and 21 conjunctival swabs. Twelve calves tested positive at both locations, while seven were positive rectally only and nine were positive only at the conjunctival site. Rectal loads ranged from 4.7 to 9,521.8 noc/μl, with ten values >100 noc/μl and five values >1000 noc/μl of which three were negative at the next sampling timepoint. The other two were negative after six and nine months. Conjunctival loads ranged from 0.8 to 37,743 noc/μl having the widest range of copy loads. 4/21 conjunctival samples showed >100 noc/μl, three >1,000 noc/μl and two of them even >10,000 noc/μl (no. 5451 and 5460 at T2). These high values also decreased sharply during the following three months and were negative (no. 5451) or at a low level of 21.9 noc/μl (no. 5460) after six months. Two calves (no. 5453, 5456) showed a rectal and one calf (no. 5460) a conjunctival load >100 noc/μl during two consecutive samplings. However, in either of these cases, the load was reduced drastically on the second sample timepoint.

**Table 6. Calves positive multiple times.**

| Loads per µl in calves (R/C) | | | | | |
|---|---|---|---|---|---|
| Animal ID | T1 | T2 | T3 | T4 | T5 |
| 5447* | na[1] | -/6.3[2] | -/52.2 | -/-[3] | na |
| 5449* | na | 57.6/24.4 | 15.3/- | 8.6/56.9 | na |
| 5450* | na | 391.3/13.9 | -/1.7 | 4.7/- | na |
| 5451* | na | 8,262.1/11,102.7 | -/8.8 | 11.5/- | na |
| 5452* | na | -/44.0 | 12.7/28.1 | 9.3/- | -/- |
| 5453* | na | 9,521.8/3.1 | 139.2/- | -/- | -/- |
| 5454 | na | 27.4/11.0 | 109.2/- | na | na |
| 5456* | na | 1,610.2/8.5 | 445.1/3.1 | 10.6/0.8 | -/- |
| 5460* | na | 5,702.9/37,743.0 | -/184.5 | -/21.9 | na |
| 5466 | na | na | 370.6/3,437.2 | -/37.5 | na |
| 5467 | na | na | 7,050.6/- | -/19.5 | na |

[1] Animal was not available.

[2] Gray shading indicates a positive result in rectal and/or conjunctival (R/C) samples with corresponding *C. pecorum* load per µl.

[3] Both samples showed a negative *C. pecorum* qPCR result.

* Calves that were integrated into the beef cattle population during this study.

## Discussion

### *C. pecorum* was the only chlamydial species found in cattle from the AgroVet-Strickhof

Many European studies based on serology indicate an endemic distribution of different *Chlamydiaceae* species in cattle with prevalences ranging from 5 to 100% (Ireland 4.75% [44], Germany 19.6% [45], Italy 24% [46], Sweden 28% [47], Austria 45% [29], Switzerland 47% [10], Germany 100% [48]). However, studies that have identified the corresponding chlamydial species are less frequently published and data on *Chlamydiaceae* prevalences in healthy Swiss cattle populations is lacking to date. Out of the four *Chlamydiaceae* representatives present in cattle, *C. abortus*, *C. suis*, *C. psittaci* and *C. pecorum* [25], only the latter was found in this study. So far, the only Swiss prevalence data with direct *Chlamydiaceae* detection originated from abortion examinations and semen analysis in bulls [7, 10, 38, 49]. In the bovine abortion studies, 235 late abortions [7, 49] and 343 placenta samples [38] from cows were examined with identification of all four *Chlamydiaceae* species. However, *C. pecorum* could only be detected in one placenta sample. In the Swiss study on bull semen samples, *C. pecorum* was not found [10]. This stands in contrast to our study indicating a significantly different *Chlamydiaceae* distribution depending on the cattle population examined, the sample material and the health status of the populations. Whereas *C. pecorum* predominates in healthy Swiss cattle populations, *C. abortus* is more common in animals with fertility problems and abortions [7, 38, 49, 50]. Interestingly, in an Austrian study, *C. pecorum* was more commonly detected in 644 dairy cows (8.9% vaginally and 8.4% cervically) than *C. abortus* (0% and 1.1%) [29]. In contrast, a German study investigating vaginal swabs from 1,074 dairy cows showed that *C. psittaci* (4.9%) was the most frequently detected species followed by *C. abortus* (3.2%) whereas *C. pecorum* was the least frequently found [28]. In a study from Poland, *C. pecorum* and *C. psittaci* was completely absent, whereas *C. abortus* and *C. suis* could be detected [50]. Outside Europe, two studies from the United States of America revealed high *C. pecorum* (39% and 59%) and *C. abortus* (24% and 12%) prevalences in healthy cattle populations of different age categories [6, 30]. In contrast to other studies [3, 34], we only detected *C. pecorum* infections

and no other chlamydial species or mixed infections. The single presence of *C. pecorum* in the conjunctiva and rectum of healthy cattle is in accordance with the hypothesis that the pathogen is endemic in ruminants worldwide [2, 12, 42]. The absence of *C. abortus* [23], which is also endemic, can be explained by the disease status in small ruminants and the associated national control strategies in Switzerland. In 2022, only 46 cases in sheep and 22 cases in goats were reported throughout Switzerland, which drastically reduces the infection pressure on cattle [51]. Taken together, distribution of the individual *Chlamydiaceae* species is strongly dependent on the sample material, the selected population, the corresponding regions, and the associated conditions such as the control measurements.

## Calves showed the highest and dairy cows the lowest *C. pecorum* prevalence

In our study, calves showed the highest animal prevalences with 55–100% and beef cattle was in the mid-range with 38.1–61.5% depending on the sampling timepoint. The lowest *C. pecorum* prevalence was recorded in dairy cows with 5.2–11.4%, which is in accordance with other studies investigating dairy herds in Austria (8.4–8.9% in vaginal/cervical samples) [29] and in the USA (7.5% in vaginal swabs and milk samples) [6], but contrasts with a lower prevalence found in Sweden (4.7% in vaginal samples) [47] and particularly in Germany (0.7%) [28]. In contrast, a study from China observed a higher *C. pecorum* prevalence of 57.2% in whole blood, milk, vaginal swabs, and fecal samples [26]. However, considering that the samples from these studies were mostly taken from the reproductive tract, it is not possible to directly compare these studies with ours.

Data on younger age categories are provided by two US studies on preselected populations. One study, in which 51 heifers were sampled vaginally, showed a *C. pecorum* prevalence of 39% [30], which was also observed in our study in the corresponding age category of beef cattle. In the second study, 41 calves were sampled via vaginal, rectal, and nasal routes during the first twelve weeks of life resulting in a prevalence of 58.5%, which corresponds with our observed *C. pecorum* positivity in calves at T4 [6]. The only European study performed in Germany showed a *C. pecorum* prevalence of 100% in a pre-selected calf herd of 13 animals when testing nasal, fecal, and conjunctival swabs [20], similar to what we observed in our study at three sampling timepoints. Rectal and conjunctival swabs, as used in our study, have been rarely used in cattle for *Chlamydiaceae* or *C. pecorum* prevalence studies. In contrast, vaginal or cervical samples were mostly collected due to a pre-selection for fertility problems.

## Conjunctival swabs were more frequently positive for *C. pecorum* than rectal swabs

Across all age categories, more conjunctival (n = 151) than rectal samples (n = 65) were positive at any sampling timepoint. The absence of positive rectal samples in dairy cows was unexpected, as the gastrointestinal tract is considered the main reservoir of *C. pecorum* [12, 20, 25] and feces is the main source of transmission [52]. In a German study with bulls sampled from three anatomical sites, *C. pecorum* was most frequently detected in fecal samples [8]. Studies evaluating bovine populations using rectal and conjunctival sample material have not yet been published but a similar study was carried out in Swiss fattening pigs. In contrast to our study, the conjunctival *C. pecorum* prevalence in these pigs was lower than the rectal prevalence [53]. An Australian study investigating 76 lambs showed that the conjunctiva was more frequently positive for *C. pecorum* than the rectum as observed in our study [42]. Despite these results, the gastrointestinal tract is still recognized as the main site of chlamydial localization in sheep [12, 54]. It is possible that ocular *C. pecorum* infections and thus excretion via tear fluid play a so far underestimated role for transmission in bovines. In particular, the absence of rectal

infections in dairy cows reinforces the possible relevance of ocular infections in this age category.

## The probability of being *C. pecorum* positive correlated with age, body weight and breeding source but not with other parameters

Another observation made in this study, seen across all animals and within the beef cattle category, was the more frequent occurrence of *C. pecorum* in younger animals. The fact that cattle often undergo chlamydial infection early in life has already been published in several studies [2, 23, 24]. Calves get often infected several times and with different strains at different mucosal sites [24]. A US study showed that *C. pecorum* infections mostly occur within the first two months of life [6]. Similar observations were also made in Australian sheep: it was shown that young ewes are more often *C. pecorum*-positive than older animals [55] and young animals are the main shedders [42]. In contrast to the age distributions in farm animals, a reversed observation was made in koalas, with older animals being more frequently *C. pecorum*-positive [56].

On the study farm, AgroVet-Strickhof, it can be assumed that every bovine had contact with *C. pecorum* during its life and that *C. pecorum* was circulating in calves. One factor leading to the high prevalence in young animals is certainly the constant mixing of the herd through purchases, which continuously introduces new and immunologically naive animals into the population. Other factors, such as increased animal density or draught are also known stressors favoring chlamydial infections [25]. It would be of particular interest to find out whether infection already takes place in the calving pen or later within the first month of life in the igloos. A US study indicated that chlamydial infections do not take place until the second week of life [6]. It is suggested that calves are free of *Chlamydiaceae* during the first week and subsequently become infected as naive animals via direct contact such as mutual licking, via feces or feed [6]. Due to the lower incidence of *C. pecorum* infections in older animals, it can be hypothesized that a competent immune response is developing with increasing age. This would need additional serological examinations but no commercial *C. pecorum*-specific ELISA tests are available to date.

This study also showed a correlation between weight and positivity for *C. pecorum*. In beef cattle at T3, lighter animals were more often positive than heavier ones which is most likely attributed to a correlation between age and weight. In addition, a higher prevalence was found in self-reared beef cattle compared to the purchased ones at two sampling timepoints (T1 and T3). However, this observation can also be explained by the age of the corresponding groups. In both cases, the purchased beef cattle were about 150 days older than the self-reared cattle, which could explain the difference in prevalence. Another cause could be a high germ load in the farms calving pen or igloos, which are known risk factors for chlamydial infections [28]. Further comparisons between positivity and breed showed no correlation in any age category. In dairy cows, no correlation between *C. pecorum* positivity and lactation status as well as the number of lactations was found.

## Mean *C. pecorum* loads were higher at younger age and at the rectum

Our study showed a significantly higher rectal load in calves compared to beef cattle, both calculated with the absolute and the mean values. Due to the lack of positive rectal samples, a comparison with the dairy cows was not possible. The distribution of the conjunctival loads based on the mean values also showed the highest load in calves. A regression analysis confirmed a correlation between younger age and higher loads at both localizations when comparing the mean values. The observed distribution confirmed the hypothesis that calves were the main shedders at the AgroVet-Strickhof farm. The *C. pecorum* load decreasing with age also

supports the hypothesis that cattle develop a competent immune response as already reported in sheep [42]. In cattle, there are no comparable studies published yet, but an Australian study on sheep detected higher loads in younger animals [27]. Specifically, weaning lambs had the highest load followed by postweaning lambs and the lowest loads were found in the oldest category named pre-slaughter lambs. Another study in lambs showed an initial shedding at two months and a peak occurring at six months of age [42]. The decrease in shedding starting at the sixth month is particularly interesting, as it is associated with seroconversion, which takes place early in life. In conclusion, a similar age distribution was observed in our bovine study comparable to those in lambs with the young animals identified as the main shedders. In contrast to these studies, however, an increase in *C. pecorum* load was observed during fattening in a Swiss study on pigs [39].

In addition, higher rectal than conjunctival loads was observed in this study. Including all age categories, the mean rectal load was 1,614.1 noc/μl, whereas the mean conjunctival load was only 816.9 noc/μl. This distribution emphasizes the relevance of the less frequent rectal findings, which cause higher environmental contamination by fecal shedding. Comparable studies on cattle are not published but similar studies in fattening pigs and lambs also showed a higher rectal *C. pecorum* load compared to conjunctival samples [39, 42, 57].

A classification of all positive samples into three categories showed that only 9.4% of all sampled cattle belonged to the category of high shedders (>1,000 noc/μl). In contrast to a study in lambs with 21.1% high shedders, this proportion was much lower in our study [42]. However, more than half of the high shedding lambs in the latter study were vaginally positive whereas rectal and conjunctival high shedders were found in only 9.2% (7/76) and therefore at a similar frequency as in our study. Similar to the lamb study [42], more conjunctival (74.8%, 113/151) than rectal samples (47.7%, 31/65) fell into the category of low shedders in our study. This illustrates that only a small number of positive animals are causing most of the environmental contamination which could be reduced by detection and separation of such animals resulting in a reduction of the infection pressure on a farm.

## The duration of infection was usually short-lived in cattle, but calves often failed to eliminate *C. pecorum* within a three-month sampling interval

The majority (n = 103) of 134 positive cattle were positive at only one single sampling time-point. Hence, a short *C. pecorum* infection duration can be assumed. Calves, however, often did not eliminate the pathogen during a three-month sampling interval. Consecutive positive samples with loads of >100 noc/μl at the same anatomical site occurred in only four cases (one beef cattle and three calves) implicating that mostly younger animals were affected. In general, consecutive positivity was rarely observed in dairy cows and beef cattle and loads were usually low in these cases. This observation supports the hypothesis of a short infection duration of less than three months in these age categories. In calves, on the other hand, all multiple positive animals were positive on at least two consecutive samplings. Most calves showed the highest loads at the first sampling, which indicates that infection occurred around the time of joining the study population. Typically, the *C. pecorum* load had already decreased drastically by the time of the following sampling but was not eliminated. This incompetence of complete elimination in calves could be due to a lack of an effective immune response at a young age. As confirmed in our study, short infection duration in calves has already been observed for *C. pecorum* [6]. In the latter study, an initial two-week increase in pathogen level was observed upon infection, which subsequently resulted in a one to three-week lasting peak. Overall, this suggested an average infection duration of three to five weeks, which was shorter than our sampling interval. Under these conditions, infections could have also occurred between two

sampling timepoints in our study and thus could have remained undetected. As in cattle, mainly short-lived *C. pecorum* infections were observed in sheep. For example, in a South Australian population, *C. pecorum* prevalence decreased from 94.2% to 12.2% within ten weeks [27]. The long sampling interval and short duration of infection also raises the possibility of continuous clearance and reinfection between the sampling timepoints. This hypothesis would be plausible due to the presumed high environmental burden of *C. pecorum* in calf populations. In addition, *Chlamydiaceae* are known to survive better in dry, cold, and shaded conditions and may even survive in such an environment for several months, thus posing a constant risk of reinfection [25]. Additionally, it must be considered that *C. pecorum* can be shed intermittently [6, 20]. To obtain confidence as to whether the infections are persistent or ongoing reinfections, the sampling intervals could be shortened, or *C. pecorum* strains could be determined and compared over time.

## Conclusion

In summary, this study provided first insights into the presence of *C. pecorum* in a healthy Swiss cattle population. As a longitudinal study, the significance outweighs the more frequently conducted single point prevalence studies and provides a good basis for further projects. A *C. pecorum* strain determination and its evaluation could allow better characterization of the infections. Since this study was performed on only one farm, further studies in different regions of Switzerland are of interest to obtain more accurate information about the actual Swiss *C. pecorum* prevalence.

## Supporting information

**S1 Fig. Mean *C. pecorum* loads.** Shown are the mean (A) rectal and (B) conjunctival *C. pecorum* loads as number of copies (noc) per μl for each age category at all five sampling timepoint (logarithmic scale, mean ± standard deviation).
(TIF)

**S2 Fig. Mean *C. pecorum* loads.** Shown is the (A) rectal and (B) conjunctival *C. pecorum* load as number of copies (noc) per μl for each age category including all sampling timepoints (logarithmic scale, mean ± standard deviation). Statistical analyses were performed using the absolute loads of each respective sampling timepoint. Significant differences were represented with asterisks: Three asterisks (***) represent p-values <0.001, two asterisks (**) represent p-values <0.01 and one (*) represents p-values between 0.01 and 0.05. Non-significant values were labeled with ns.
(TIF)

**S3 Fig. Correlation between *C. pecorum* load and age including all bovines.** In this figure, the correlation between age and absolute loads are shown at the rectal (left panel) and at the conjunctival site (right panel). A regression analysis using a confidence interval of 95% was performed.
(TIF)

**S1 Table. Overview of the number of samples taken per animal.** A total of 308 bovines were sampled at least one time. The dairy herd was the most consistent age category with 34 animals tested at all five sampling timepoints.
(DOCX)

**S2 Table. Sample sizes.** Overview of the sample sizes for each age category at all five sampling timepoints.
(DOCX)

**S3 Table. Primer and Probe.** Summary of primer and probe used for qPCR (*Chlamydiaceae* and *C. pecorum*) as well as for the internal amplification control.
(DOCX)

**S4 Table. Limit of detection (LOD).** LOD performed with the wildtype *C. pecorum* strain "W73" and with the *C. pecorum* standard used in this study.
(DOCX)

**S5 Table. Comparison of different prevalences between age categories.** P-values for comparison of different prevalences between age categories are shown. The age categories are abbreviated as d (dairy cows), b (beef cattle) and c (calves). Comparisons were considered significant if the p-value was < 0.05.
(DOCX)

**S6 Table. Comparison of different prevalences with age.** P-values for comparisons between age and positivity for animal and sample prevalence are shown. For these calculations all age categories were included evaluated at all five sampling timepoints. Comparisons were considered significant if the p-value was < 0.05.
(DOCX)

**S7 Table. Comparisons of animal prevalences with age within each age category.** P-values for comparisons of animal prevalences with age within each age category and sampling timepoint. Comparisons were considered significant if the p-value was < 0.05.
(DOCX)

**S8 Table. Comparisons between *C. pecorum* positivity rates with breed, lactation status and lactation number.** P-values for comparisons between *C. pecorum* positivity rates (animal prevalence) with breed, lactation status and lactation number at each sampling timepoint. Comparison with breed was performed in each age category whereas the other comparisons were only performed in dairy cows. Comparisons were considered significant if the p-value was < 0.05.
(DOCX)

**S9 Table. Mean *C. pecorum* loads per µl.** Listed are the mean *C. pecorum* loads per µl with corresponding standard deviation (SD) for each age category, anatomical site and at each sampling timepoint.
(DOCX)

**S10 Table. Correlation between the *C. pecorum* load and age.** P-values of the correlation between the *C. pecorum* load and age are shown. Absolute numbers of copies (noc) as well as mean values were used for calculations in each age category and including all bovines. Comparisons were considered significant if the p-value was < 0.05.
(DOCX)

**S11 Table. Comparison between the *C. pecorum* load and the anatomical localization.** P-values of the comparison between the *C. pecorum* load (absolute numbers as well mean values) and the anatomical localization within each age category and including all bovines are shown. Comparisons were considered significant if the p-value was < 0.05.
(DOCX)

**S1 Text. Regression analysis.** To test for a correlation between age and loads, a regression analysis was performed across all animals calculated with the following formula: Loads = a + bx.
(DOCX)

## Acknowledgments

The authors are grateful to Barbara Prähauser, Institute of Veterinary Pathology, Vetsuisse Faculty, University of Zurich, for the support with the laboratory work. We further thank Mirjam Klöppel, the head of the animal caretake team at the AgroVet-Strickhof, and her dedicated team. They played a crucial role in helping with the sampling process. We thank Dr. Melissa Terranova and Dr. Wolfgang Pendl, research coordinators at the AgroVet-Strickhof, for their help in the coordination of the project.

## Author Contributions

**Conceptualization:** Samuel Loehrer, Nicole Borel.

**Data curation:** Hanna Marti, Michael Hässig.

**Formal analysis:** Samuel Loehrer, Hanna Marti, Michael Hässig.

**Funding acquisition:** Nicole Borel.

**Investigation:** Samuel Loehrer, Fabian Hagenbuch, Hanna Marti, Michael Hässig, Nicole Borel.

**Methodology:** Samuel Loehrer, Fabian Hagenbuch, Hanna Marti, Theresa Pesch, Michael Hässig.

**Project administration:** Hanna Marti, Michael Hässig, Nicole Borel.

**Resources:** Nicole Borel.

**Supervision:** Hanna Marti, Michael Hässig, Nicole Borel.

**Validation:** Samuel Loehrer, Hanna Marti, Theresa Pesch, Michael Hässig, Nicole Borel.

**Visualization:** Samuel Loehrer, Hanna Marti, Nicole Borel.

**Writing – original draft:** Samuel Loehrer, Fabian Hagenbuch, Nicole Borel.

**Writing – review & editing:** Samuel Loehrer, Fabian Hagenbuch, Hanna Marti, Theresa Pesch, Michael Hässig.

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
