## [Decision Letter · Decision Letter 0]

18 Oct 2023

PONE-D-23-30712Longitudinal study of Chlamydia pecorum in a healthy Swiss cattle populationPLOS ONE

Dear Dr. Borel,

Thank you for submitting your manuscript to PLOS ONE. After careful consideration, we feel that it has merit but does not fully meet PLOS ONE’s publication criteria as it currently stands. Therefore, we invite you to submit a revised version of the manuscript that addresses the points raised during the review process. Editor's comments: Please reduce the size of the manuscript, especially the Discussion, which is exceedingly long. Also, please edit the manuscript as per the reviewer's detailed suggestions. A point by point response is required to adequately assess the revised manuscript.

We look forward to receiving your revised manuscript.

Kind regards,

Deborah Dean, M.D., M.P.H.

Academic Editor

PLOS ONE

Journal Requirements:

This study was supported by Seed Money AgroVet-Strickhof from the Vetsuisse Faculty and the Fondation sur la Croix.

The authors are grateful to Barbara Prähauser, Institute of Veterinary Pathology, Vetsuisse faculty, University of Zurich, for the support with the laboratory work. We further thank Mirjam Klöppel, the head of the animal caretake team at the AgroVet-Strickhof, and her dedicated team. They played a crucial role in helping with the sampling process. We thank Dr. Melissa Terranova and Dr. Wolfgang Pendl, research coordinators at the AgroVet-Strickhof, for their help in the coordination of the project. This research was supported by the AgroVet-Strickhof research facility, a cooperation between Strickhof, ETH Zurich and University of Zurich.

This study was supported by Seed Money AgroVet-Strickhof from the Vetsuisse Faculty and the Fondation sur la Croix.

Reviewers' comments:

Reviewer's Responses to Questions

**Comments to the Author**

1. Is the manuscript technically sound, and do the data support the conclusions?

Reviewer #1: Yes

2. Has the statistical analysis been performed appropriately and rigorously? 

Reviewer #1: Yes

3. Have the authors made all data underlying the findings in their manuscript fully available?

Reviewer #1: Yes

4. Is the manuscript presented in an intelligible fashion and written in standard English?

Reviewer #1: Yes

5. Review Comments to the Author

Reviewer #1: This study by Loehrer et al focusses on determining the prevalence of Chlamydia pecorum in a Swiss cattle population of dairy cows, beef cattle and calves. Chlamydia pecorum causes a broad range of diseases/conditions in small and large ruminants worldwide and is probably the most common chlamydial infection in these host animals. This is a very neglected area of Chlamydia research and an area that we have very little data on globally and particularly in Europe. The manuscript is very well written and the research and analyses are very well conducted. The findings support those in similar studies in the US and elsewhere and add very important information on the situation in Switzerland. It was very interesting to see how sampling location and age affected prevallence, as well as the differences between the three categories of bovines. It would of course also be very interesting to know the situation elswhere in Europe to compare to the situation in Switzerland, and hopefully this study will spark similar investigations elsewhere.

My main crticism is that the manscript is unnecessarily long and can and should be cut back in length. But other than that other comments are very minor with a number of similar typographical errors throughout. Once these are corrected and the Discussion cut back I would recommend publication of this manuscript. The work does represent a very important addition to the literature on this neglected area of chlamydial research.

Minor comments:

1. Line 231. Add a reference or details of the 76 bp MOMP sequence.

Moderate comments:

1. There is so much in depth detail given in the manuscript that at times it feels like there is just too much information. The manuscript would be improved by condensing some of this detail down to the pertinent main results.

3. The Discussion is extremely long at over 10 pages in length and reads more like a review than a Discussion of the main results in the context of the literature. There are parts that can and should be cut back. For example, "The lowest C. pecorum prevalence was recorded in dairy cows with 5.2%-11.4%, which is in accordance with other studies investigating dairy herds. For example, in a study from Austria, a randomized dairy herd (n = 644) was sampled and C. pecorum prevalences of 8.9% (vaginal) and 8.4% (cervical) were obtained [29]. Another study with similar prevalences as in our study was conducted in the USA where vaginal swabs and additionally milk samples from dairy cows were evaluated resulting in a prevalence of 7.5% [6]. In contrast to our study, a lower prevalence was recorded in Sweden where only 4.7% of pre-selected dairy cows showed a positive C. pecorum result in vaginal swabs [47]. A randomized German study (n = 1’074) measured an even lower C. pecorum prevalence of only 0.7% [28]." This could be cut back to something like "The lowest C. pecorum prevalence was recorded in dairy cows with 5.2-11.4%, which is in accordance with other studies investigating dairy herds in Austria (8.4-8.9% in vaginal/cervical samples) [29] and in the USA (7.5% in vaginals swabs and milk samples) [6], but contrasts with a lower prevalence found in Sweden (4.7% in vaginal samples) and particulalry in Germany (0.7%) [28]. This is just one example, but illustrates the point. I think the Discussion could easily be cut back by 4 pages by making such changes to reduce the comparative detail which is not necessary.

Minor typos/corrections:

1. Lines 7/9. Add superscripts to affiliation linkages.

2. Line 31. Use of "north-eastern" versus "northeastern" elsewhere (eg line 113). Be consistent

3. Line 34. Instead of "localisation" I would suggest using "sampled area".

4. Line 37. Instead of "included" I suggest "were determined to be". Also in % ranges only include the "%" at the end of the range ie 5-10% not 5%-10%. Same point for lines 662/664/665 for example.

5. Line 61. Place a comma after "species".

6. Line 62. Instead of ".... pigs), wild....." put "..... pigs) and wild....".

7. Line 75. Insert "a" before "herd level"

8. Line 82. Should be "cows".

9. Line 87. Replace "...“Buss-Encephalitis” later...." with ".....“Buss-Encephalitis”, which was later.....".

10. Line 98. "Microarray" should not be capitalised.

11. Line 134. "10'029 l/cow/year" I would suggest writing as "10,029 L/cow/year".

12. Line 173. Replace "....period whereas 202 of them were tested at least twice, ....." with ".....period, with 202 of them tested at least twice, .....".

13. Line 174. Replace "....in both localisations...." with "....from both sampled areas.....".

14. Line 174. Add a comma after "In particular, ".

15. Line 211. Add a full-stop after "...(Table 1)."

16. Line 226. Add a hyphen to "seven-fold".

17. Line 300. "1’564" should be "1,564".

18. Line 304. Replace “localization” with “location” or "area".

19. Lines 301/2. Commas after both "(Fig1A)" and "(Fig1C)".

20. Line 349. "the localization" I suggest rewording to "their sampled location".

21. Line 362. I suggest making the title of Fig 2 more informative. "Fig 2. Anatomical localization" doesnt really describe what the Figure is about.

22. Line 370. I suggest replacing "localisation" with "location".

23. Line 455. " 2’743.7 noc/µl" should be " 2,743.7 noc/µl".

24. Line 457. "1’739.7 noc/µl" should be "1,739.7 noc/µl".

25. Line 517. Same point as above re "34’531.9 noc/µl" and "0.8-37’743 noc/µl". Also lines 573/594/597/598/801/808. Also correct in all Tables.

26. Line 522/571/592. "localizations" should be replaced with "locations"

27. Line 692. Should "either" be "both"?

28. Line 859. I suggest replacing "Synopsis" with "Conclusion"

6. PLOS authors have the option to publish the peer review history of their article (what does this mean?). If published, this will include your full peer review and any attached files.

Reviewer #1: No

---

## [Author Response · Author response to Decision Letter 0]

21 Nov 2023

Resubmission of manuscript PONE-D-23-30712

Dear Dr. Dean,

Thank you for giving us the opportunity to submit a revised version of the manuscript PONE-D-23-30712 by S. Loehrer et al. entitled „Longitudinal study of Chlamydia pecorum in a healthy Swiss cattle population” to PLOS ONE.

Enclosed you will find the point-by-point response to the comments of the editor and the reviewer (in italics) as well as the revised manuscript.

Yours sincerely,

Nicole Borel

Corresponding author:

Nicole Borel, Prof., FVH Pathology, Dipl. ECVP

Institute of Veterinary Pathology, Vetsuisse Faculty

University of Zurich, Winterthurerstrasse 268

CH-8057 Zurich

Switzerland

Tel.: ++41 44 63 58563

Fax.: ++41 44 63 58934

Email: nicole.borel@uzh.ch

 

Journal Requirements:

The revised manuscript meets PLOS ONE's style requirements.

Thanks for this information, we have deposited the manuscript on BioArchive.

We do not have specific grant numbers for the awards.

This study was supported by Seed Money AgroVet-Strickhof from the Vetsuisse Faculty and the Fondation sur la Croix.

We received funding from the two organizations above (without grant numbers). The Seed Money AgroVet-Strickhof was internal (our organization, the Vetuisse Faculty). The Funding from the Fondation sur la Croix was external.

The authors are grateful to Barbara Prähauser, Institute of Veterinary Pathology, Vetsuisse faculty, University of Zurich, for the support with the laboratory work. We further thank Mirjam Klöppel, the head of the animal caretake team at the AgroVet-Strickhof, and her dedicated team. They played a crucial role in helping with the sampling process. We thank Dr. Melissa Terranova and Dr. Wolfgang Pendl, research coordinators at the AgroVet-Strickhof, for their help in the coordination of the project. This research was supported by the AgroVet-Strickhof research facility, a cooperation between Strickhof, ETH Zurich and University of Zurich.

This study was supported by Seed Money AgroVet-Strickhof from the Vetsuisse Faculty and the Fondation sur la Croix.

The Acknowledgments section as above does not contain funding. The support mentioned in the Acknowledgments section was not of financial nature.

The ORCID ID has been validated.

The reference list is complete and correct. It does not contain any retracted articles.

Response to the Editor and Reviewer 1:

Editor’s comments:

Please reduce the size of the manuscript, especially the Discussion, which is exceedingly long. Also, please edit the manuscript as per the reviewer's detailed suggestions. A point by point response is required to adequately assess the revised manuscript.

As suggested, we have reduced the size of the manuscript, especially the Discussion. The manuscript has been edited according to the reviewer's detailed suggestions.

Reviewer 1:

This study by Loehrer et al focusses on determining the prevalence of Chlamydia pecorum in a Swiss cattle population of dairy cows, beef cattle and calves. Chlamydia pecorum causes a broad range of diseases/conditions in small and large ruminants worldwide and is probably the most common chlamydial infection in these host animals. This is a very neglected area of Chlamydia research and an area that we have very little data on globally and particularly in Europe. The manuscript is very well written and the research and analyses are very well conducted. The findings support those in similar studies in the US and elsewhere and add very important information on the situation in Switzerland. It was very interesting to see how sampling location and age affected prevallence, as well as the differences between the three categories of bovines. It would of course also be very interesting to know the situation elswhere in Europe to compare to the situation in Switzerland, and hopefully this study will spark similar investigations elsewhere.

We thank this reviewer for the positive feedback.

My main criticism is that the manscript is unnecessarily long and can and should be cut back in length. But other than that other comments are very minor with a number of similar typographical errors throughout. Once these are corrected and the Discussion cut back I would recommend publication of this manuscript. The work does represent a very important addition to the literature on this neglected area of chlamydial research.

As suggested, we have reduced the size of the manuscript, especially the Discussion. All typographical errors have been corrected as suggested.

Minor comments:

1. Line 231. Add a reference or details of the 76 bp MOMP sequence.

The reference Pantchev et al. has been added.

Moderate comments:

1. There is so much in depth detail given in the manuscript that at times it feels like there is just too much information. The manuscript would be improved by condensing some of this detail down to the pertinent main results.

We have condensed the manuscript as suggested.

3. The Discussion is extremely long at over 10 pages in length and reads more like a review than a Discussion of the main results in the context of the literature. There are parts that can and should be cut back. For example, "The lowest C. pecorum prevalence was recorded in dairy cows with 5.2%-11.4%, which is in accordance with other studies investigating dairy herds. For example, in a study from Austria, a randomized dairy herd (n = 644) was sampled and C. pecorum prevalences of 8.9% (vaginal) and 8.4% (cervical) were obtained [29]. Another study with similar prevalences as in our study was conducted in the USA where vaginal swabs and additionally milk samples from dairy cows were evaluated resulting in a prevalence of 7.5% [6]. In contrast to our study, a lower prevalence was recorded in Sweden where only 4.7% of pre-selected dairy cows showed a positive C. pecorum result in vaginal swabs [47]. A randomized German study (n = 1’074) measured an even lower C. pecorum prevalence of only 0.7% [28]." This could be cut back to something like "The lowest C. pecorum prevalence was recorded in dairy cows with 5.2-11.4%, which is in accordance with other studies investigating dairy herds in Austria (8.4-8.9% in vaginal/cervical samples) [29] and in the USA (7.5% in vaginals swabs and milk samples) [6], but contrasts with a lower prevalence found in Sweden (4.7% in vaginal samples) and particulalry in Germany (0.7%) [28]. This is just one example, but illustrates the point. I think the Discussion could easily be cut back by 4 pages by making such changes to reduce the comparative detail which is not necessary.

We have reduced the Discussion by implementing the suggested wording above and by further reduction of the text.

Minor typos/corrections:

1. Lines 7/9. Add superscripts to affiliation linkages.

2. Line 31. Use of "north-eastern" versus "northeastern" elsewhere (eg line 113). Be consistent

3. Line 34. Instead of "localisation" I would suggest using "sampled area".

4. Line 37. Instead of "included" I suggest "were determined to be". Also in % ranges only include the "%" at the end of the range ie 5-10% not 5%-10%. Same point for lines 662/664/665 for example.

5. Line 61. Place a comma after "species".

6. Line 62. Instead of ".... pigs), wild....." put "..... pigs) and wild....".

7. Line 75. Insert "a" before "herd level"

8. Line 82. Should be "cows".

9. Line 87. Replace "...“Buss-Encephalitis” later...." with ".....“Buss-Encephalitis”, which was later.....".

10. Line 98. "Microarray" should not be capitalised.

11. Line 134. "10'029 l/cow/year" I would suggest writing as "10,029 L/cow/year".

12. Line 173. Replace "....period whereas 202 of them were tested at least twice, ....." with ".....period, with 202 of them tested at least twice, .....".

13. Line 174. Replace "....in both localisations...." with "....from both sampled areas.....".

14. Line 174. Add a comma after "In particular, ".

15. Line 211. Add a full-stop after "...(Table 1)."

16. Line 226. Add a hyphen to "seven-fold".

17. Line 300. "1’564" should be "1,564".

18. Line 304. Replace “localization” with “location” or "area".

19. Lines 301/2. Commas after both "(Fig1A)" and "(Fig1C)".

20. Line 349. "the localization" I suggest rewording to "their sampled location".

21. Line 362. I suggest making the title of Fig 2 more informative. "Fig 2. Anatomical localization" doesnt really describe what the Figure is about.

22. Line 370. I suggest replacing "localisation" with "location".

23. Line 455. " 2’743.7 noc/µl" should be " 2,743.7 noc/µl".

24. Line 457. "1’739.7 noc/µl" should be "1,739.7 noc/µl".

25. Line 517. Same point as above re "34’531.9 noc/µl" and "0.8-37’743 noc/µl". Also lines 573/594/597/598/801/808. Also correct in all Tables.

26. Line 522/571/592. "localizations" should be replaced with "locations"

27. Line 692. Should "either" be "both"?

28. Line 859. I suggest replacing "Synopsis" with "Conclusion"

All typos have been corrected and minor changes were made as

---

## [Editor Report · Decision Letter 1]

27 Nov 2023

Longitudinal study of Chlamydia pecorum in a healthy Swiss cattle population

PONE-D-23-30712R1

Dear Dr. Borel,

We’re pleased to inform you that your manuscript has been judged scientifically suitable for publication and will be formally accepted for publication once it meets all outstanding technical requirements.

Kind regards,

Deborah Dean, M.D., M.P.H.

Academic Editor

PLOS ONE
---

## [Editor Report · Acceptance letter]

1 Dec 2023

PONE-D-23-30712R1 

Longitudinal study of *Chlamydia pecorum* in a healthy Swiss cattle population 

Dear Dr. Borel:

I'm pleased to inform you that your manuscript has been deemed suitable for publication in PLOS ONE. Congratulations! Your manuscript is now with our production department. 

Kind regards, 

on behalf of

Dr. Deborah Dean 

Academic Editor

PLOS ONE